# New Ceramic Tiles Produced Using Old Technology Applied on Historic Roofs—Possibilities and Challenges

**DOI:** 10.3390/ma15217835

**Published:** 2022-11-06

**Authors:** Krzysztof Ałykow, Łukasz Bednarz, Magdalena Piechówka-Mielnik, Magdalena Napiórkowska-Ałykow, Michał Krupa

**Affiliations:** 1Team of Civil Engineers, 59-800 Luban, Poland; 2Faculty of Civil Engineering, Department of Building Structures, Wrocław University of Science and Technology, 50-370 Wroclaw, Poland; 3Faculty of Architecture, Chair of Housing Environment, Cracow University Technology, 31-155 Cracow, Poland

**Keywords:** ceramic tiles, historic roofs, old technology, conservation, heritage

## Abstract

In the case of historic buildings, especially those under protection, it is important to replace elements of the roof covering, while maintaining current technical standards, to meet the requirements of the conservator. The authors of the article present alternatives to commonly used solutions, based on their experience with replacing historic building roofing with ceramic tiles made according to the production and firing technology of the nineteenth century. They emphasize that the correct/specialized restoration of existing tiles in a building makes it possible to preserve and reuse them, which is in line with the principles of historic preservation. However, due to the preservation of the roof tiles, it is not always possible to revitalize them. As a solution to the problem, the use of clay roof tiles manufactured according to 19th-century firing technology, including handmade methods, is presented, which preserves the geometry of the historic roof tiles. The approach presented by the authors meets both the requirements of conservation theory and the building standards for roofing elements. Although it is much more expensive than the solutions currently commonly used that result from modern technical requirements and tile-manufacturing technology, in the case of objects of significant cultural heritage, it is a solution that meets modern technical requirements while not compromising the original appearance of the monument.

## 1. Introduction

The frequent replacement of old elements with new elements in historic buildings (such as ceilings, roof trusses, window and door joinery, plaster, roof coverings, insulation, the body of the walls, etc.) irretrievably deprives them of their historical and scientific value, causing the problem that after a thorough “revitalization”, what remains is a “candy” new building that merely imitates the original one. This is not only contrary to the principles of protection that constitute the rule of law [1,2,3], but it deprives us of a material legacy that is a visible sign of the activity and presence of the generations before us.

In particular, when it comes to roofs, it is common to approach original tiles as merely a technological element that protects the roof of the building from weather, which is usually understandable, but in the case of historic buildings covered with historic handmade tiles, it is most often not correct [4,5,6]. Therefore, if the value of a monument is determined by its originality, why are the original roofing materials removed in most cases and replaced with modern machine-made ones?

This paper presents the authors′ proposal on how to simultaneously meet the technical criteria arising from contemporary technical standards while at the same time satisfying the requirements arising from the theory of historic preservation. This is of great importance, especially in the case of buildings of significant historical and artistic value and those under the protection of the Office for the Protection of Monuments.

From the point of view of the ceramic tile roofing technology commonly available on the market and widely used today, it would seem impossible to produce a roof covering manufactured in accordance with modern building standards while at the same time meeting the requirements of monument conservation theory. Therefore, ensuring that the proper requirements of modern building standards are met while ensuring authenticity and preserving the traces of historic alterations and transformations is an often-overlooked aspect of the activities of today′s architects and engineers.

The basic classification of roof tiles is based on the material from which they are made, that is, ceramic tiles (fired from clay) and cement tiles (made from a combination of cement and sand with additives). Ceramic tiles are slightly lighter than cement tiles and are available in a wider range of colors. However, the natural red brick shade is the most popular because it blends best with a variety of facades and surrounding areas.

Of today′s clay roof tiles, the most common shapes used are the Dutch (also known as S-shaped), overlapping, Marseille, or plain tile (Figure 1). Flat roof tiles are available as clay and cement tiles. A modern solution is the photovoltaic tile, which makes it possible to produce electricity. This type of tile is a novelty on the construction market, but is not recognized by the conservation community.

Over the years, as a result of UV radiation, rain, and other atmospheric conditions, tiles can become stained, lose color, and thus start to look unattractive (Figure 2a,b). To renovate the tiles, it is not necessary to replace the entire roof immediately. It is also possible to renovate them, waterproof the surface, and strengthen the structure with suitable chemicals based on silane and silicate particles.

The most effective method is to clean the tiles using a pressure washer (Figure 3a) with a rotary nozzle at a pressure of min. 220 bar. Using this method, loose dirt, moss, and old, poorly adhering coatings can be removed. Particular care should be taken to thoroughly remove moss from the tiles.

The most common means of restoring historic roof tiles after cleaning are silicate-based agents (Figure 3b), which do not add a new color to the ceramics (they are colorless), thus achieving colorless protection while at the same time impregnating and closing the pores in the old tiles. This action also protects them from moisture. This is particularly important under the influence of changing weather conditions. By closing the pores in the ceramics, the silicates create a smoother finish and thus inhibit moss growth on the roof. Tile waterproofing treatments are available in gloss, satin, and matt finishes. Unfortunately, not all roof tiles can be revitalized, especially if their structure and mechanical properties disqualify them.

As the problem of preserving old tiles is very complex and costly, the most common route is to choose a new material to mimic the old one. The use of new machine-made roofing tiles has many negative consequences:Changes to the appearance of the roof covering;The obliteration of the original arrangement of tiles, often of different shapes, built into the roof slope and constituting evidence of its transformation relevant from the point of view of monument documentation;The necessity of leveling the roof slope for new tiling, resulting in the incorporation or, worse, the removal of original carpentry elements or their fragments;The need to apply new layers, including vapor-permeable foil, which sometimes leads to a disruption of the microclimate within the loft and acceleration of the biological corrosion process of the original elements of the roof trusses;A reduction in the weight of the roof covering, which in the case of high roofs (e.g., churches) and in places of contact with high partitions (e.g., church towers) often results in the tiles being torn out of the roof slope despite their proper fixing.

For roof coverings of historic buildings, it is important to produce adequate documentation that includes measurements of the geometry of the existing tiles and, if necessary, their layout on the slope of the roof. An example of such measurements of the geometry of historic roof tiles to help produce replacements is shown in Figure 4.

It is equally important to determine the degree of technical degradation of historic roofs and, in the case of undamaged roof tiles, how they should be cleaned, maintained, and possibly reinforced. The next step is to indicate the extent and manner of replacing the roofing elements.

Tiles that are part of the restoration and replacements of original ones should be adapted in a way that does not obliterate the original architectural layout and made in such a way as to meet the technical requirements, while allowing specialists to recognize the original and secondary elements, according to the requirements of monument conservation.

According to the authors, one of the most interesting solutions is the use of ceramic tiles made according to 19th-century firing technology, hand-formed and fired in traditional coal-fired Hoffman furnaces.

Based on the results of the laboratory tests carried out by the authors and on the authors′ experience with repairs/maintenance of the roof coverings of historical buildings, the article shows that solutions are possible and available that meet both technical and conservation criteria at the same time. This is very important in order to avoid falsifying the historical appearance of a building, which is unacceptable from the point of view of heritage protection.

## 2. Materials and Methods

To compare the physical and mechanical performance of old and new tiles that look the same, we tested the methods for making historic clay roof tiles, dated as min. 100 years old. The historic tiles were compared with new tiles made with traditional technology, but with up-to-date technical parameters. The tests included determining water absorption and permeability, assessing frost resistance, and a bending strength test. There were 52 tiles used in the tests, of which one was damaged during transport—sample No. 051 cracked.

Before testing, the material was visually assessed. It was found that:The tiles came in various sizes and shapes;The age of the tiles was estimated to be around 100 years;Traces of various mortars were observed on many tiles;The tiles were heavily and moderately soiled;Numerous roof tiles were chipped;There were numerous traces of biological corrosion on the right (upper) surfaces of the tiles, which was contributed to by moss and algae growth (visible, for example, on sample Nos. 010, 029, and 038);One of the tiles (sample No. 051) was cracked.

### 2.1. Water Absorption Test

A total of 30 samples of the dismantled clay tiles (No. 021 to No. 046, No. 048 to No. 050, and No. 052) were selected from the tiles supplied for the test, dried for 24 h in an oven at 110 °C +/− 5 °C (Figure 5a), and weighed to the nearest 1 g. Then, they were subjected to a water absorption test (Figure 5b) according to the procedure in Appendix A of the current standard in [7]. The results of the n_m_ mass absorption were between n_m min_ = 9.7% (sample No. 031) and n_m max_ = 15.4% (sample No. 029), as shown in Figure 6. The average absorption for all 30 tiles was 13.9%.

Unfortunately, there are no standards for the maximum water absorption values of clay plain tiles. According to the world′s operating major tile manufacturers, the maximum water absorption of approved clay roof tiles should not exceed 10%. This water absorption condition was not met by 28 of the 30 samples.

### 2.2. Frost Resistance Test

The frost resistance test was performed according to the current standard in [8]. Before the frost resistance test, 6 tiles (Nos. 021, 024, 029, 031, 038, and 050) were selected from the 30 tiles (No. 021 to No. 046, No. 048 to No. 050, and No. 052; tile No. 047 was a broken tile) on which the tile water absorption test was performed. Tiles that were free from unacceptable damage in the test and characterized by their minimum, maximum, and average water absorption (2 pieces each) were selected and subjected to frost resistance tests according to the procedure presented in the standard in [8] (Figure 7). Table 1 summarizes and compares the tiles before and after the frost resistance test.

The following were observed on tiles subjected to the frost resistance test after the test: surface scratches, peeling, delamination, and spalling, examples of which are shown in Figure 8a–c.

According to the standard in [9], the clay roof tiles used in Central Europe should be Class 1; that is, they should not show any of the types of damage specified in Table 1 of the standard in [8] after 150 freeze/thaw cycles. The condition of resistance to frost was not fulfilled.

### 2.3. Permeability Test

The permeability test was performed according to the current standard in [10]. After selecting 10 samples (No. 011 to No. 020) of tiles, the tests were carried out according to the procedure described in the standard in [10] (Figure 9). The tile permeability test consisted of determining the time that elapsed until the first drop of water fell under the pressure of the water column exerted on the upper surface of the tile, under normal atmospheric conditions. The test consisted of lying the tile samples in tap water at room temperature for 48 h ± 4 h. The samples were then dried at 110 °C ± 5 °C to a constant weight. The final step was to cool for 4 h at room temperature. The test lasted no longer than 20 h. The permeation coefficient (IC) was calculated using the relevant formulae.

The test results for the individual tiles and the time to the first drop of water (h) are shown in Figure 10. The average time to the first drop of water for the entire test series was X_i av_ = 4.98 h. The highest value of the single water absorption coefficient was ICX_i av_ = 0.751. The mean permeation coefficient was ICX_i av_ = 0.751. The value of the largest single-sample permeation coefficient was ICX_i_ = 0.988.

According to the standard in [9], for standard clay tiles approved for Category 1, the average permeability coefficient of ICX_i av_ should be 0.8 or less, and all values of the permeability coefficient of ICXi for individual samples should be 0.85 or less. The condition for the average permeation coefficient ICX_i av_ was met. The condition for the values of the permeation coefficients for individual ICX_i_ samples was not met by seven out of ten samples.

### 2.4. Flexural Load Capacity Test

The bending resistance test was carried out according to the current standard in [7]. After selecting 10 clay tile specimens (No. 001 to No. 010), the tests were carried out according to the procedure described in the standard in [7] (Figure 11a,b). Figure 12 summarizes the load-bearing results obtained on the selected clay roof tiles tested. The average failure load of F_av_ was 0.24 kN.

According to [9], for approved plain clay tiles, the bending strength should take a value of min. 600 N (0.60 kN). The bending strength condition was not met in all 10 samples.

In comparison, hand-formed tiles and fired according to the technology of the nineteenth century in coal-fired Hoffman furnaces have the properties shown in Table 2 and meet the requirements of the current standards [7,8,9,10].

## 3. Results of Case Study

The laboratory tests carried out for historic clay tiles showed that the roofing material did not meet the modern requirements for reuse in the renovated building. Basic physical–mechanical tests reflected the condition of the historic tiles as a material unsuitable for effective roofing, such as:The absorbability (water absorption) condition was not met by 28 out of 30 samples;Although the condition regarding the average water absorption coefficient ICX_iav_ was met, the value of the water absorption coefficient for single samples, ICX_i_, was not met by 7 out of 10 samples;The condition concerning frost resistance was not fulfilled;The condition concerning flexural capacity was not fulfilled.

In comparison, new tiles formed by hand and fired according to 19th-century technology in coal-fired Hoffman furnaces, although almost identical to the historic tiles, complied with the current standards (Table 2).

To illustrate the applicability of new ceramic tiles produced with old technology, a short case study of the application of this type of tile in a historic building is presented below. The reference object is the Salt House (Figure 13) of 1539, located in southern Poland [11]. It is located in a space between two rows of medieval defense walls erected before 1220. In 1566, there was a fire in the town, but the building was not damaged besides a few wooden elements. Extended in 1698, the building was used as a storage house for salt and grain until the end of the 18th century. In the 19th century, it was partly converted for use as a prison and the headquarters of the town′s fire brigade. The building served as the firemen′s headquarters up to the 1990s of the twentieth century.

The Salt House is situated on a hill within the chamber of the ramparts, i.e., between the higher and lower lines of the ramparts. This location was due to fire safety considerations, since the high walls effectively separated the building from the flammable buildings of the city, mostly wooden, and also protected it from flooding in the event of rainfall and floods.

The Salt House was built on a rectangular plan measuring 33.8 × 18.4 m from basalt stone and field pebbles bonded with lime mortar with a clay mixture, with the north wall being coextensive with the inner fortification wall. Its numerous but small window openings on the southwest, south, and southeast sides of the town could be used as rifle ranges if necessary. It eventually reached a height of 22 m, accommodating four stories and an attic.

Inside, all of the ceilings received a wooden structure and were supported by wooden columns. The different levels were connected by stairs in the form of ladders supported by ceiling beams. Inside the warehouse, a crane was installed to allow for the loading or unloading of goods. Originally, there must have also been a writer′s room and a weighing scale on the ground floor level. The main stone-vaulted gate was located on the northeast side of the building [12,13,14].

The project documentation created for the renovation envisaged the removal of the existing 19th-century machine-made tiles of no historical or technical value and their replacement with new clay roof tiles, with the application of contemporary layers used in the laying of the new roof, including vapor-permeable foil, which is a commonly used solution [15], but not always correct.

Once the work had started, it was found that on the roof, in addition to tiles dating back to the nineteenth century produced by machine in an amount of approximately 25%, the majority of the remaining were handmade tiles, including some from the period of the building′s construction, with four different modes of shaping (flat, segmental, semicircular, and angular) (Figure 14a).

It was also found that on the south slope, the tiles were laid in a scalloped pattern (on the north slope in a lace pattern) with lime mortar fixings with fur and the joints sealed with wooden shackles.

Following the intervention of the building inspector after the renovation work had already begun, the voivodeship conservator of monuments changed his earlier decision to remove the tiles in their entirety and replace them with new ones. He ordered the careful removal, under the additional supervision of a conservator of works of art, of about half of the tiles on the southern slope, ordering that the tiles fixed with original mortar be left on the roof. After analyzing the problem, it was decided to undertake the conservation of the original roofing elements in situ using a hoist and climbing methods (on ropes), while reinforcing the slope by introducing additional battens supporting the original ones. This allowed the original patches and mortar to be preserved and prevented possible damage to the original tiles during their removal and reinstallation after conservation. Only selected roof tiles in poor condition were replaced (Figure 14b). These measures were taken after taking into account all the conservation principles of this type of historic structure [16], while monitoring the static condition of the structure. Some of the procedures proposed in [17] were implemented.

The original decision of the preservation office was also modified, ordering that the original existing roof tiles be cleaned, preserved, and incorporated using traditional mortar fixing. Not all tiles were suitable to undergo this process due to their technical condition.

Instead of the heavily damaged original and 19th-century machine-made tiles, custom handmade tiles resembling the original in appearance, but with much better physical and mechanical properties and with a detailed arrangement of tiles of different shapes on the roof slope reflecting its historic character, were installed (Figure 15).

## 4. Discussion

Investigations of historic clay roof tiles have shown significant losses in the quality of the material as a roof covering used in traditional construction. Their failure to meet the requirements for water absorption, permeability, and frost resistance results in poor durability of the material, compared to tiles produced today. Their strengthening through conservation measures, i.e., cleaning and then structural strengthening with chemical preparations based on silicates, is possible, as in the case of the case study presented above, but in the opinion of the authors, due to both the significantly higher costs of this type of work in comparison to ordinary roofing and the greater technological complexity requiring the participation of conservators of works of art in the field of ceramics, it is justified only in the case of historic buildings of particular artistic, scientific, or historical significance. For historic buildings that are not of such significance, but are important from the point of view of individual cultural heritage protection or the protection of a larger urban or rural area [18,19], for example, for the preservation of Genius Loci, the solution proposed by the authors meets the criteria of cultural heritage protection, whose importance is emphasized in the document of the International Council of Monuments and Sites [20].

The use of contemporary handmade roofing tiles, fired in traditional coal-fired Hoffman furnaces, which give the shape, size, and color of historic tiles, makes it possible to achieve a quality roof covering. The importance of this problem can be learned from numerous publications [21,22,23,24].

In the case of the presented case study, the change in the method of investment, i.e., the abandonment of the originally designed contemporary tiles in favor of handmade ceramic tiles made according to 19th-century technology in coal-fired Hoffmann cookers and the in situ conservation of part of the original tiles, resulted in an extension of the investment from one to three years and a significant threefold-increase in costs. However, because of a different method of revitalizing the roof covering of the building compared to that originally envisaged, it was possible to preserve the historic elements of the roof. This was achieved in compliance with both contemporary technical requirements and the requirements of the conservation doctrine, thus precisely complying with the requirements recommended by UNESCO [25] for the protection of the historic urban landscape.

This is particularly evident where, along with the original, cleaned, and structurally reinforced roofing elements, new ones have been built in, but produced according to a model and technology corresponding to the historic technology. On the one hand, this approach meets the technical requirements of contemporary standards, while on the other hand, it does not obliterate the original historic appearance of the monument by introducing modern elements that do not harmonize with the original historic substance. An additional and perhaps the greatest benefit of the approach proposed by the authors is that, despite the coherent and harmonious appearance of the historic building′s revitalized roofing, the use of 19th-century tiles makes it possible and, more importantly, will in the future enable specialists to identify and differentiate between original and contemporary elements. In the opinion of the authors, this approach also perfectly fulfills the authenticity document requirements of the Nara [26] for the preservation of the originality of historical elements.

## 5. Conclusions

In the opinion of the authors, the use of tiles formed by hand and fired according to 19th-century technology in coal-fired Hoffman kilns not only meets the technical and conservation requirements in the case of historic buildings, but also prevents many of the disadvantages that accompany the replacement of roofing with new, machine-made contemporary roofing. The tiles, which are a precise reproduction of the geometry, texture, and color of the original roofing elements, are in keeping with the conservation doctrine of preserving the traces of our ancestors′ activities and presence as accurately as possible. They do not introduce cognitive dissonance when viewing the monument, as is the case with additions or the replacement of original elements with completely new ones, but only supplement or finally replace the destroyed elements in a way that allows the monument to be fully perceived in its original form without falsifying its history.

There are doubts about the cost of making new handmade tiles or maintaining old tiles compared to using contemporary materials. The question is: Is it easier and quicker to replace the roof covering with a new one made of commonly available materials? While preserving for posterity the authenticity of heritage buildings, together with their scientific and historical significance, and fulfilling the requirements of conservation doctrine, we should revitalize the roof coverings of historic buildings in a way that allows the building′s unadulterated past to be maintained. This choice should not be questioned.

## Figures and Tables

**Figure 1 materials-15-07835-f001:**
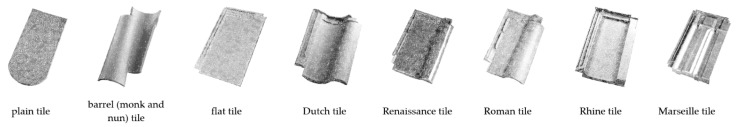
Examples of different roof tiles.

**Figure 2 materials-15-07835-f002:**
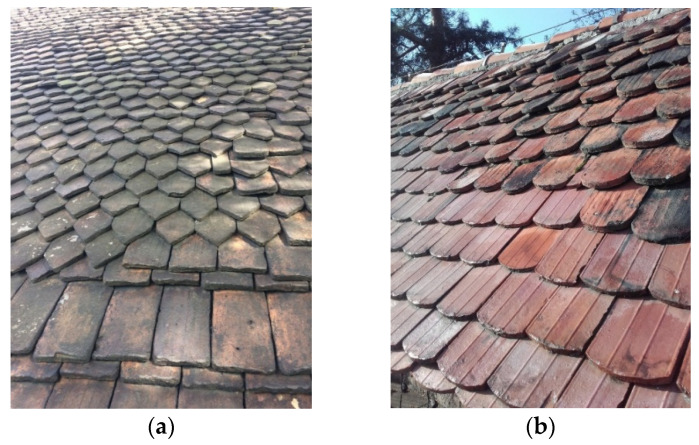
Examples of historic roof coverings after many years of use: (**a**) handmade late medieval and Renaissance roofing tile; (**b**) machine-made tiles approximately 100 years old.

**Figure 3 materials-15-07835-f003:**
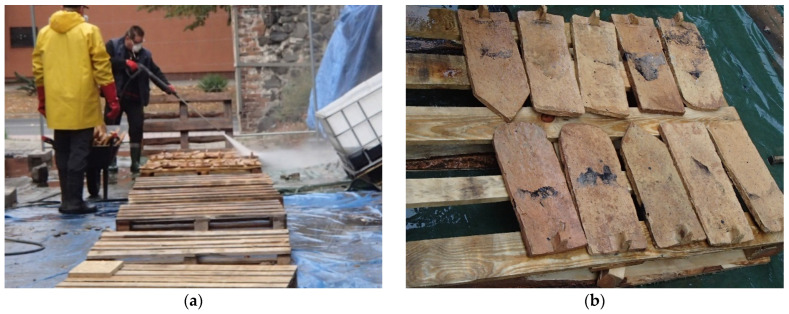
Revitalization of historic roof tiles: (**a**) cleaning; (**b**) maintenance.

**Figure 4 materials-15-07835-f004:**
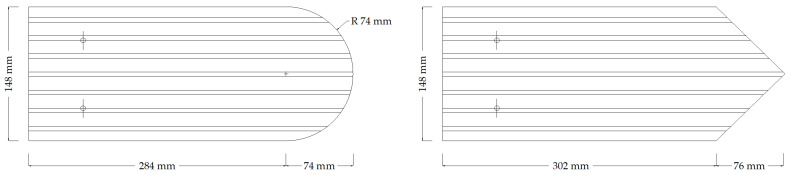
Geometry of historic roof tiles.

**Figure 5 materials-15-07835-f005:**
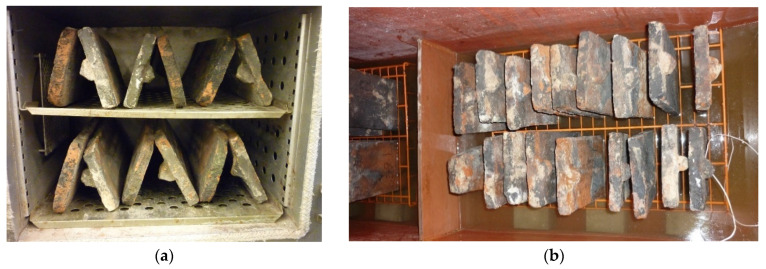
Absorption test: (**a**) drying of tiles to constant weight at a temperature of 110 °C +/− 5 °C; (**b**) roof tiles in a chamber with 100% humidity.

**Figure 6 materials-15-07835-f006:**
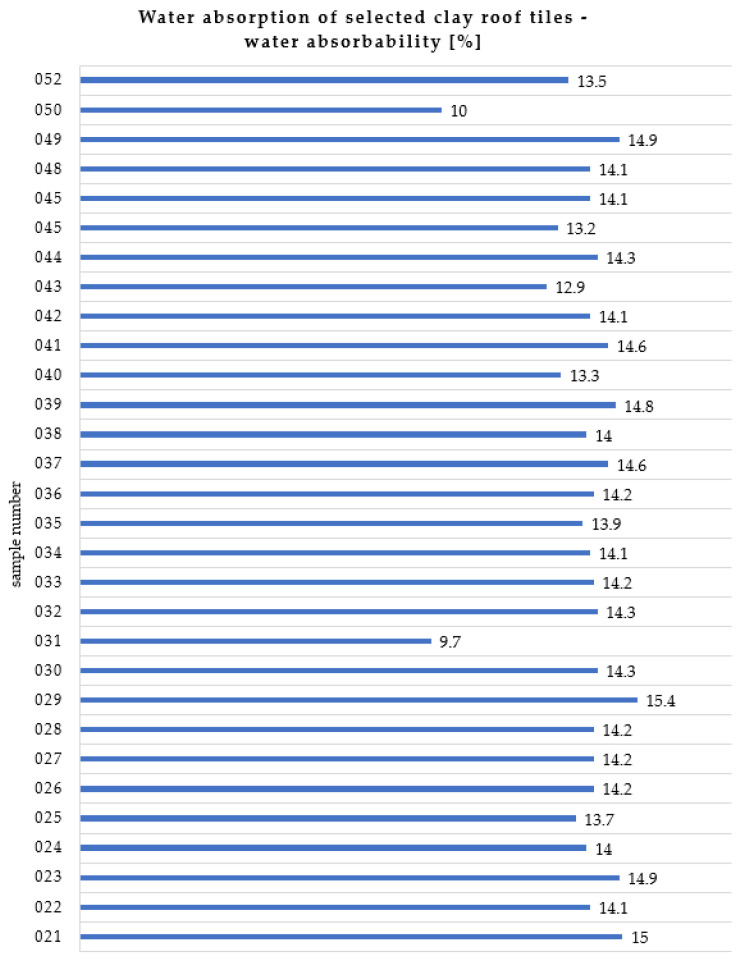
Water absorption of selected historical clay roof tiles.

**Figure 7 materials-15-07835-f007:**
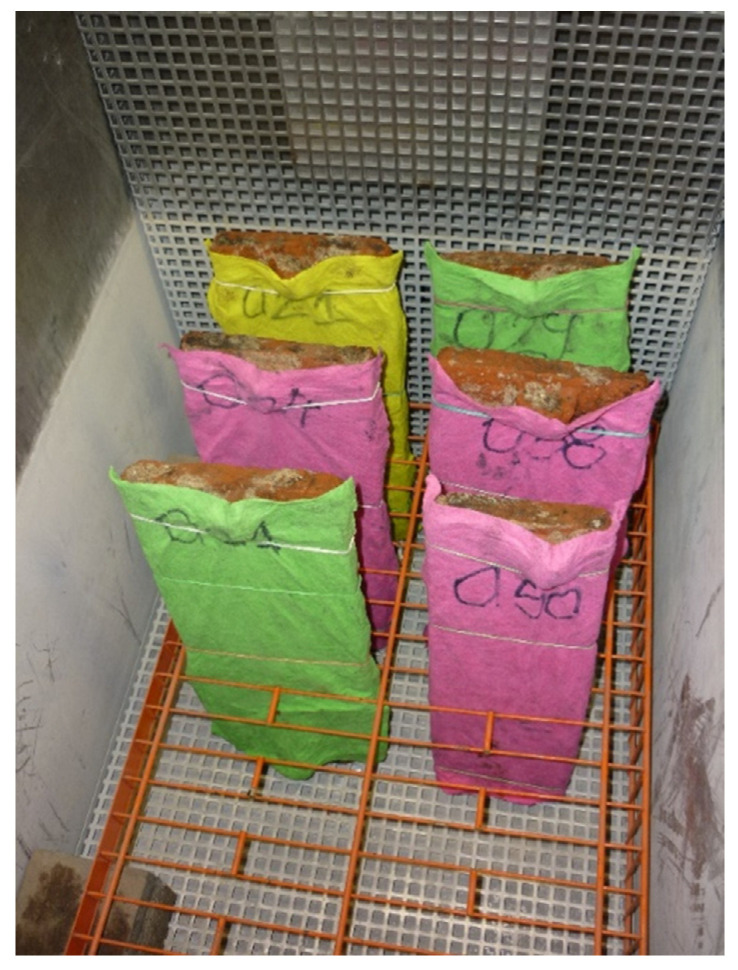
Roof tiles in a climate chamber.

**Figure 8 materials-15-07835-f008:**
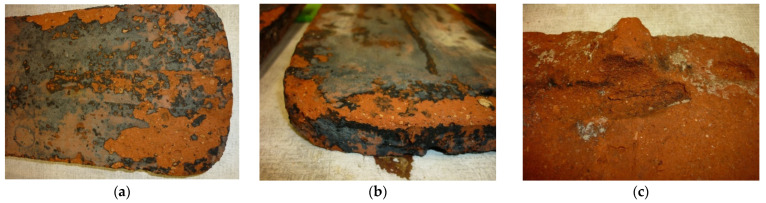
Frost resistance test: (**a**) surface damage observed after the test; (**b**) edge damage observed after the test; and (**c**) damage observed after the test.

**Figure 9 materials-15-07835-f009:**
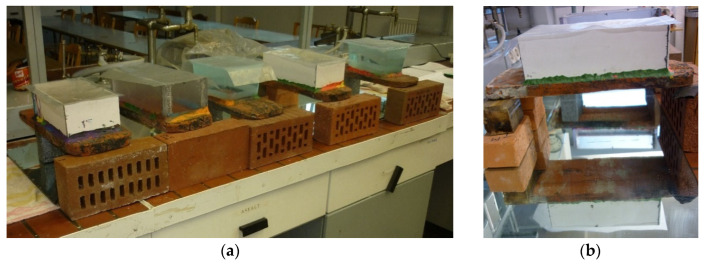
Roof tiles during the impermeability test: (**a**) general view; (**b**) view of the underside of tile.

**Figure 10 materials-15-07835-f010:**
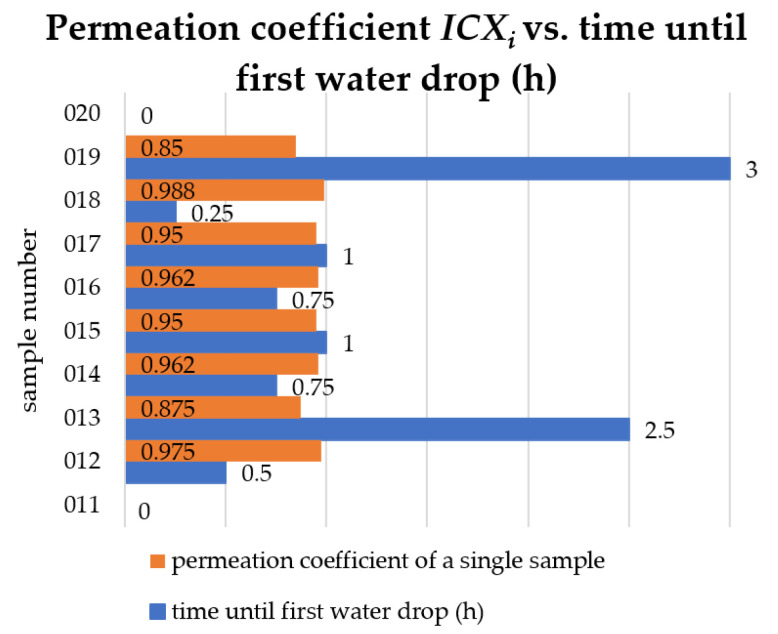
Permeation coefficient.

**Figure 11 materials-15-07835-f011:**
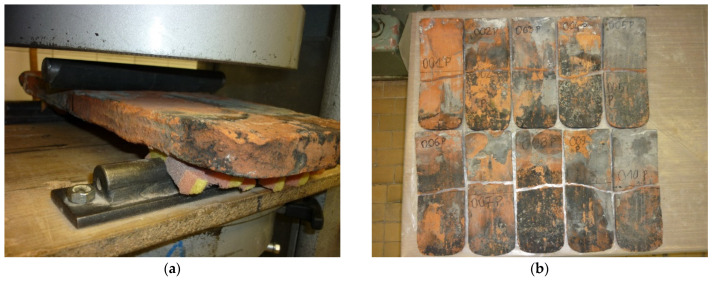
Flexural load capacity test: (**a**) tile in the testing machine during flexural capacity tests; (**b**) tiles (Nos. 001–010) after flexural strength tests.

**Figure 12 materials-15-07835-f012:**
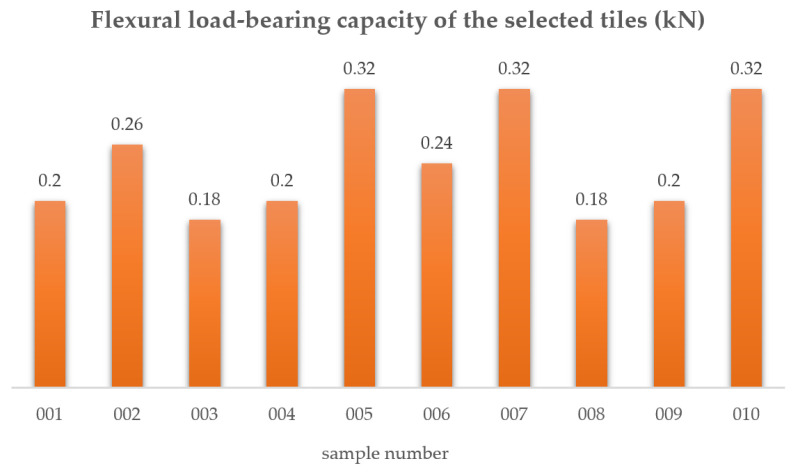
Flexural load-bearing capacity of the selected tiles.

**Figure 13 materials-15-07835-f013:**
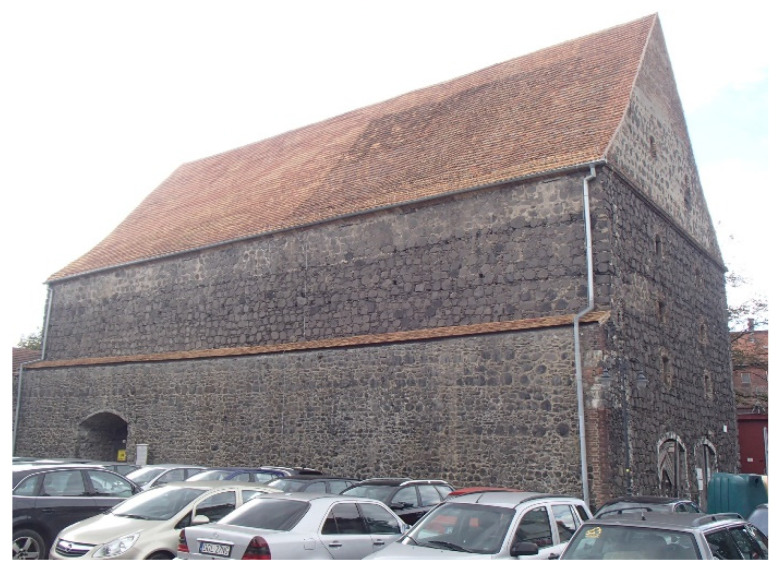
The Salt House elevation view.

**Figure 14 materials-15-07835-f014:**
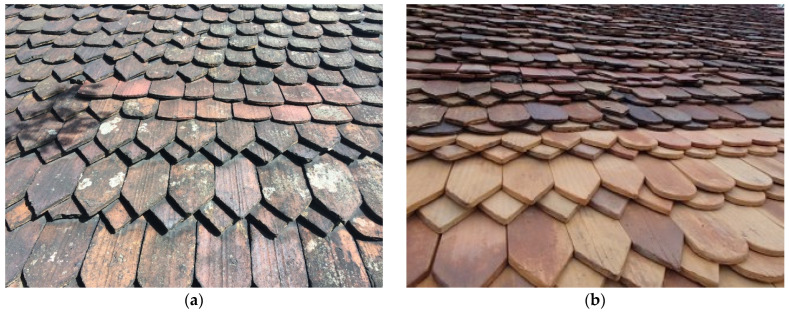
Different types of roof tiles: (**a**) historical; (**b)** new roof tiles in historic dimensions.

**Figure 15 materials-15-07835-f015:**
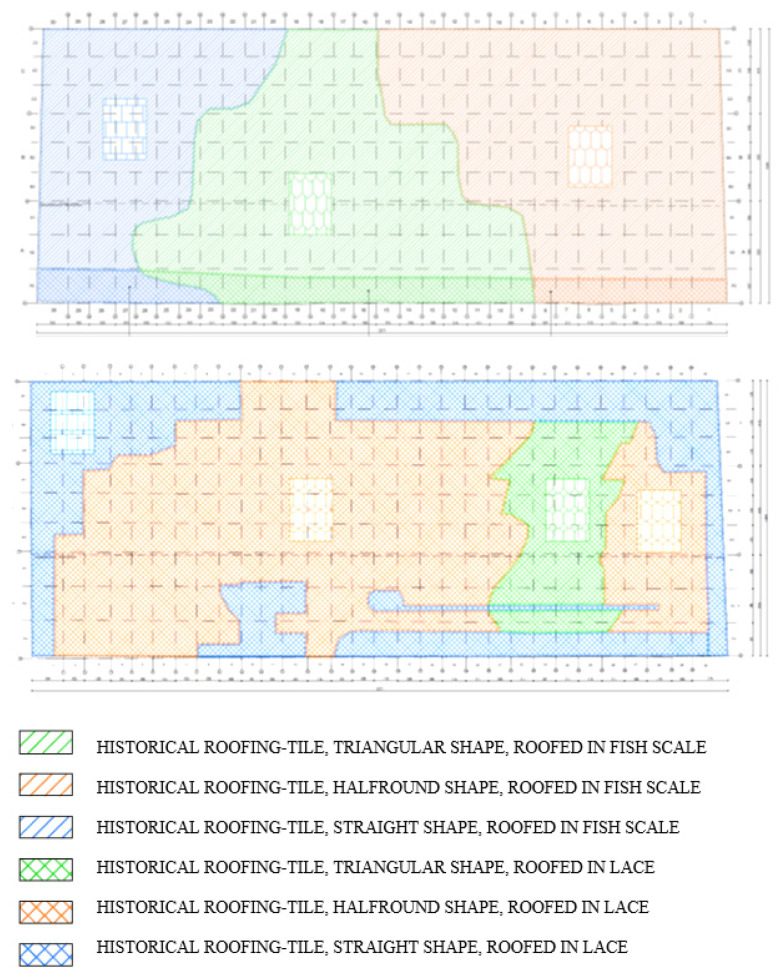
Arrangement of tiles on roof slopes.

**Table 1 materials-15-07835-t001:** Comparison of the visual condition of the selected tiles before and after the frost resistance test.

Tile No.	Appearance Before Frost Resistance Testing	Appearance After Frost Resistance Testing
021	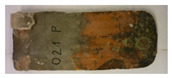	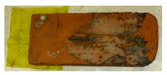
024	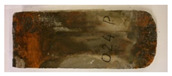	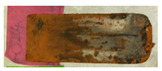
029	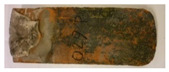	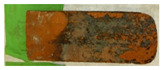
031	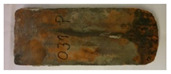	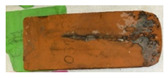
038	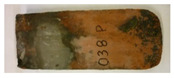	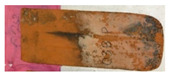
050	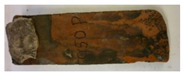	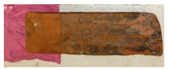

**Table 2 materials-15-07835-t002:** Physical and mechanical properties of the new ceramic tiles produced with the old technology.

Properties	Value
Water absorption	<10%
Permeability	ICx_i_ ≤ 0.85
Frost resistance	Frost resistant after 150 cycles
Flexural load capacity	≥1.2 kN

## Data Availability

Not applicable.

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
