# Peer review of "New Ceramic Tiles Produced Using Old Technology Applied on Historic Roofs—Possibilities and Challenges"

_materials, 2022, doi:10.3390/ma15217835_

Round 1
Reviewer 1 Report
A scientific paper about this topic seems to be important.
However, i suggest to illustrate the different types of tiles schematically to allow easy insight for also a non-domain-professional reader.
Also the paper should be read differently than a "construction instruction" ("Ensure a stable and clean substrate before applying them.")
Fig 3 is of rather low quality / resolution, please improve.
Fig 5 as such is too small. also - it might be wise to somehow show the differences between different roof clay tiles.
Fig 9 has numbers within the bars, which is difficult to read - please improve.
Please avoid 3d-bars (Figure 11).
Somehow the paper is everything and nothing, as you bring some general information on the one hand, and a small case study. The conclusions are really poor - what do you want to say? Yes tile retrofit and cleaning is an alternative to new tiles. However, what does it cost, what is the effort, what are the shortcomings.
Bring in a "Future Research" section (e.g. what happens if some of the tiles can t be used and need to be replaced/reproduced?). and a "Limitations of this study" section as well.
Reviewer 2 Report
In my opinion manuscript materials-1966287 is well written and deserves publication after revision.
Some suggestion to improve the manuscript
1. At the end of the abstract add the novelty of the work.
2. Introduce more literature references on ceramic tile roof conservation. In the end of the introduction: explain originality of the work, indicate work hypothesis, aim and objectives of the paper.
3. Organize better the manuscript.
Section 2 should contain only the enumeration of materials and methods used. The results of the tests performed should be given in the Results and Discussion section. Compare your results with others from similar interventions.
What were the criteria to replace some of the tiles?
What about trying to consolidate the original tiles by impregnation? Discuss also this option in the text.
4. Correct typos, revise English, for example:
- Avoid repetitions at line 74 “negative consequences” appears twice
- Create bullet lists at lines 76 – 87, lines 113-122
Round 2
Reviewer 1 Report
the authors properly adressed all relevant aspects. as such, i think the paper shall be published!
Reviewer 2 Report
In my opinion the revised manuscript materials-1966287 meets the publication standards of Materials journal.